# Overemphasis on recovery inhibits community transformation and creates resilience traps

Benjamin Rachunok [1,4✉] & Roshanak Nateghi [1,2,3]

Building community resilience in the face of climate disasters is critical to achieving a sustainable future. Operational approaches to resilience favor systems' agile return to the status quo following a disruption. Here, we show that an overemphasis on recovery without accounting for transformation entrenches 'resilience traps'–risk factors within a community that are predictive of recovery, but inhibit transformation. By quantifying resilience including both recovery and transformation, we identify risk factors which catalyze or inhibit transformation in a case study of community resilience in Florida during Hurricane Michael in 2018. We find that risk factors such as housing tenure, income inequality, and internet access have the capability to trigger transformation. Additionally, we find that 55% of key predictors of recovery are potential resilience traps, including factors related to poverty, ethnicity and mobility. Finally, we discuss maladaptation which could occur as a result of disaster policies which emphasize resilience traps.

[1] School of Industrial Engineering, Purdue University, West Lafayette, IN, USA. [2] Purdue Climate Change Research Center, Purdue University, West Lafayette, IN, USA. [3] Center for the Environment, Purdue University, West Lafayette, IN, USA. [4] Present address: Department of Civil & Environmental Engineering, Stanford University, Stanford, CA, USA. ✉email: rachunok@stanford.edu

There has been a significant increase in the frequency of major disasters costing over $1 billion USD, with the direct costs of disasters during 2018–2019 exceeding $136 billion USD[1]. Accelerated urbanization, aging infrastructure, climate change and reactive federal disaster policies that prioritize recovery over mitigation and thus incentivize development in high-risk areas have amplified the vulnerability of communities to climate disasters[2].

Resilience has long served as an organizing principle for marshalling resources and managing government and private sector investments to reduce vulnerability and stimulate recovery in response to major natural hazards and disruptions[3,4]. Theoretical and analytical studies of resilience exist in the social sciences, ecology, urban planning, and engineering[3,5]. Despite disciplinary differences, resilience is broadly conceptualized as capacities to bounce back after shocks and systematically adapt and transform to preserve system functionality[6–8].

More recently, there is a push to move beyond ontological discussions of resilience towards an operational paradigm at the community level[9,10], with a community understood as geographically linked groups of interacting individuals with shared norms and interests[11]. Despite recent advancements in operational models of community resilience[11–14], fundamental knowledge gaps remain. These gaps can be traced to the overwhelming focus of operational models of resilience on bouncing back[15] after disruptions and thus preserving the status quo. Specifically, in existing paradigms, a resilient system deviates minimally from its current state and returns to the status quo rapidly upon disruption[3,4,16–23,23–25]. The focus on recovery is referred to as 'engineering resilience'[26,27], and this paradigm has served as the foundation for decision and policymaking frameworks aimed at building resilient and sustainable systems and communities[19,28–34].

At their core, the engineering resilience frameworks quantify how communities are disrupted and recover—typically through measuring reliable access to critical infrastructure such as the electric power grid—and seek to identify risk factors within communities and/or systems which promote a rapid return to pre-disruption states. While these approaches are beneficial for prioritizing relief and recovery efforts, they fall short in fully operationalizing resilience in a way which also promotes mitigation, adaptation, and transformation. Moreover, these frameworks may be promoting maladaptation—defined as actions that are beneficial in the short term but ultimately increase vulnerability to future disruptions[35,36]. There is evidence that current disaster policies based on engineering resilience paradigms—such as insurance and disaster relief assistance programs—exacerbate wealth inequality[2] and broaden the racial wealth gap in areas impacted by disasters[37] by reinforcing the status quo which exacerbates persistent vulnerability rather than enabling adaptation[36]. Operationalizing resilience paradigms that incentivize not only recovery but also transformation will enable designing disaster policies and interventions which do not exacerbate vulnerabilities and inequities.

In this work, we shrink the gap between the concept and operationalization of resilience by quantifying both recovery and communities' transformation. Defining transformation as a 'systemic change of the urban system'[8] which includes nonlinear reorganizations of infrastructure, ecosystems, lifestyles, institutions, and governance[8,38,39], we measure and track the reorganization and transformation of communities in addition to quantifying their recovery from disruption. Specifically, we leverage the state of the art in statistical machine learning to (i) establish key predictors of recovery, and (ii) identify which of these key risk factors are conducive to catalyzing or inhibiting transformation. We quantify threshold effects and conduct tipping point analyses by estimating the degree of change needed in risk factors to cause transformation, using the 2018 Hurricane Michael in Florida as a case study. The focus on hurricanes was due to the severity of their impact on communities and their wide-reaching devastation, positioning them as significant stress tests of community resilience. According to the National Oceanic and Atmospheric Administration, hurricanes have caused more deaths and destruction than any other recorded climate disaster in U.S. history. Hurricane Michael was specifically chosen as a case study as it is the most powerful storm to make landfall in Florida since the state began publicly reporting county-level disaster impacts. As of 2018, 41% of hurricanes that have hit the US have made landfall in Florida[40], and accordingly, the state has a significant number of programs designed to foster resilience and aid in the immediate recovery from hurricanes. We use power outages—available at a county level throughout the storm—as a proxy for community recovery, as power outages were widespread throughout the state affecting communities' access to all other critical resources such as food, water, transport, and hygiene. Our findings demonstrate that an overemphasis on recovery and not accounting for transformation can entrench resilience traps, where risk factors that are predictive of recovery inhibit positive transformation and perpetuate maladaptive states.

## Results

**Quantifying recovery**. State-of-the-art approaches for measuring the recovery of a community utilize predictive models to relate risk factors to disaster outcomes as access to critical services is interrupted and restored[11,18,41–44]. Given the localized and place-based nature of community resilience, and in line with the previous studies[12,14,45], we perform predictive modeling of recovery at a county level. We first identify key risk factors that are predictive of recovery and then calculate their relative contribution to recovery across all counties in the state of Florida. In other words, we consider a large pool of county-level risk factors related to the environmental opinions, sociodemographic, economic, housing, and mobility characteristics for each of the 67 counties in Florida[46,47] which encompass many of the common indices utilized in measuring social vulnerability (See Supplementary Table 1 for a list of the risk factors, their sources, and descriptions). The risk factors serve as independent variables in an ensemble-of-trees predictive model (see "Methods") of restored access to electricity—used here as a proxy for recovery—after 2018 Hurricane Michael in Florida, while controlling for population and hazard exposure (Fig. 1b). We select a subset of 20 risk factors which are most predictive of restored access to electricity (Fig. 1), using an ensemble-of-trees predictive model. Specifically, using an exhaustive search approach, our three-stage variable selection algorithm (see "Methods") searches through all possible combination of input variables and identifies the smallest subset of risk factors that are most predictive of community recovery (see "Methods"). These risk factors represent what a data-driven approach to engineering resilience would identify as most important for restoring access to critical services.

**Quantifying transformation**. To quantify how communities transform, we develop a new approach termed Contrastive Community Networks (CCN) (Box 1). CCN is grounded in Self-Organized Maps (SOMs), a class of unsupervised learning techniques for simultaneous dimension reduction and projection[48]. The CCN utilizes a SOM to create a relational network of communities—here counties in Florida (Fig. 2a)—in which proximity in the network corresponds with similarity in the portfolio of risk factors between counties (see "Methods").

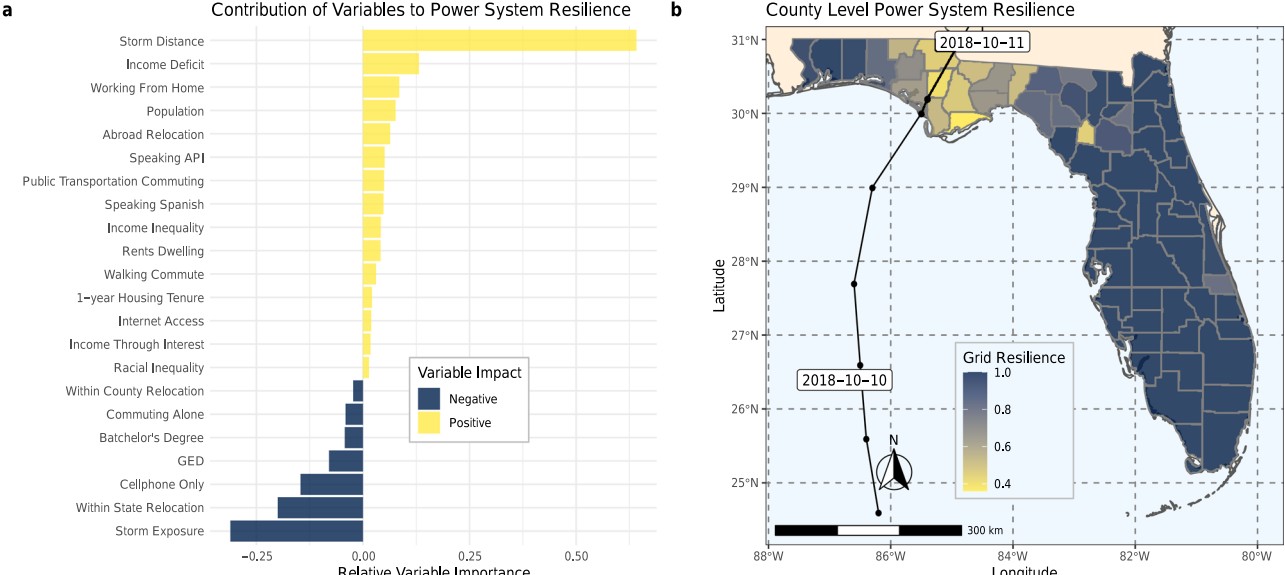

**Fig. 1 Risk factors contributing to recovery of engineered systems (i.e., restored access to electricity). a** Shows which risk factors have the highest relative contribution to recovery and whether they make a positive (yellow) or negative (blue) contribution to restored access to electricity; and (**b**) a map colored by the 'engineering resilience' of the power grid to Hurricane Michael along with the storm's track. Darker counties were more resilient to the storm. Figures (**a**, **b**) created in R (v 3.2.1; https://www.r-project.org/)[88] using the ggplot2 package (v 3.3.0; https://ggplot2.tidyverse.org/)[102]. Plot (**b**) additionally used usmap (v 0.5.0; https://github.com/pdil/usmap)[103]. Map shapefiles in b are from usmap and the US Census Bureau[104].

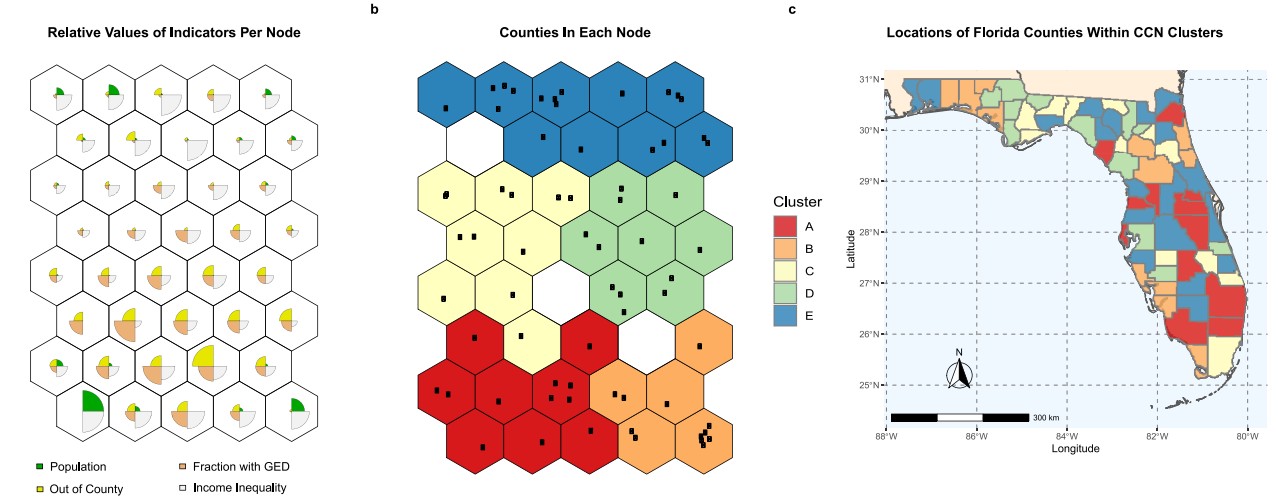

**Fig. 2 Illustration of the baseline CCN for county clustering. a** Values of four sample community risk factors across each CCN node. The wedges within each node correspond to the relative ranking of the risk factor in each node, shown here to illustrate how the CCN nodes capture similarity across dimension. **b** Mapping of each county within the CCN. Black dots represent a county being mapped to that node within the CCN (c) Location of CCN clusters from (**b**) (colors) within Florida. Each county is colored with the color of the node it is assigned to in (**b**). Figures (**a**–**c**) created in R (v 3.2.1; https://www.r-project.org/)[88] using the ggplot2 (v 3.3.0; https://ggplot2.tidyverse.org/)[102] and kohonen (v 3.0.10)[98]. Figure (**c**) additionally used usmap (v 0.5.0; https://github.com/pdil/usmap)[103]. Map shapefiles in (**b**) are from usmap and the US Census Bureau[104].

In contrast to previous methods which track temporal trends of composite risk/vulnerability or resilience indices as a proxy for community change[14,49,50], the CCN algorithm measures transformation by detecting changes in risk factors substantial enough to shift a county's position in the relational network. Thus, rather than relying on an individual (opaque) composite index, we quantify transformation by measuring the degree of contrast between a county and its peers while considering the entire portfolio of risk factors. The 20 risk factors identified by the predictive model (aka 'the engineering resilience model') as contributing the most to recovery (Fig. 1) are used as inputs to

the CCN to create a 'baseline': i.e., to establish the network of similarities between the communities against which transformation will be measured. In this step, 48 input nodes are selected to form the baseline CCN (see "Methods") and each county is mapped to one of the CCN nodes based on the values of the risk factors (Fig. 2b). Counties which occupy adjacent or nearby nodes in the CCN have greater similarity in the 20 risk factors (Fig. 2a). As communities transform and their risk factor values change, their similarity with others will morph; resulting in a reconfiguration of the CCN and subsequently a county being mapped to an alternative node in the CCN.

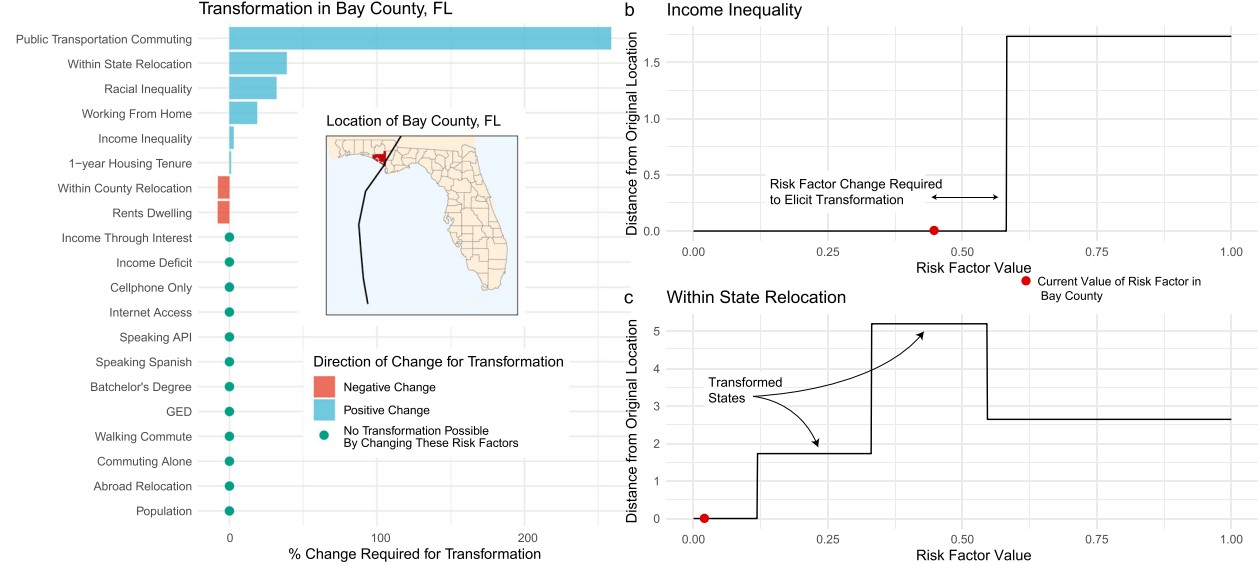

**Fig. 3 Community change required for transformation. a** The amount of change required to elicit transformation in Bay County, FL for each risk factor. Blue indicates a positive risk factor shift is required for transformation, while red indicates a negative shift, and green dots show community risk factors for which no transformation is possible. **b**, **c** As each community risk factor is perturbed ($x$-axis of **b**, **c**), we compute where Bay County is mapped in the CCN. The values of the risk factors occurring at each jump indicate the degree of risk factor perturbation leading to a reconfiguring of the CCN. (**b**) shows the change for income inequality, (**c**) for within state relocation.

**Transformation trajectories and thresholds**. After creating a baseline CCN, we perturb each risk factor for each county, monitoring the configuration of the CCN at every updated value. We do this until the perturbation is great enough such that the structure of the CCN re-organizes or 'tips' into an alternative configuration and county is mapped to a non-baseline node (Fig. 3b, c). We use a county being mapped to a non-baseline node as an indicator of transformation. In other words, in the CCNs architecture, transformation occurs when a change in the risk factors of a county is large enough, such that the county's position in the relational network shifts and it becomes more similar to an alternate set of peers from its baseline. By defining transformation relative to the peers of a county, we avoid imposing judgment about the magnitude of change in a risk factor significant enough to be identified as transformation.

Tracing the location of a county within the CCN as it re-organizes is called the county's transformation trajectory[51,52] (see Supplementary Fig. S3), and the distance from the original to updated node within the CCN corresponds with the degree of transformation experienced (see Methods). To illustrate the insights that can be drawn from this approach coupled with the engineering resilience (aka recovery) model, we calculate the temporal trajectories for each risk factor in Bay County Florida (Fig. 3): a county which experienced extensive damage due to Hurricane Michael[53].

Results indicate that in Bay County, only 8 risk factors (40% of those evaluated) have the possibility of triggering transformation (Fig. 3a). Stated alternatively, improving these risk factors will enhance both the immediate disaster recovery of the county—as determined by the recovery model—and will potentially change the county's similarity to its peers. Changes in the remaining 60% of the risk factors, however, can only improve the recovery of the community, and would not alter the underlying similarity of the county to its peers (i.e., not conducive to transformation).

For those which trigger transformation, we define a transformation threshold: the percentage increase or decrease in the risk factor associated with CCN reconfiguration. We conceptualize a transformation threshold as the magnitude of change in a risk factor required to trigger transformation. The aim is to jointly identify the risk factors within a community which are conducive to triggering transformation, and determine the degree of change needed in these risk factors for transformation to occur. Transformation thresholds provide a relative comparison of the importance of risk factors as they contribute to transformation, such that county-level risk factors with lower thresholds are more conducive to transformation. In Bay County, two risk factors— the county-level fraction of individuals who moved within the county and county-level fraction of renter-occupied housing— have negative transformation thresholds while the other six (Fig. 3a) are positive. The six positive risk factors are the county-level fraction of the population commuting primarily by public transportation, the county-level fraction of the population who has moved to a given county from elsewhere in Florida in the past year, the county-level measures of racial and income inequality, the county-level fraction of the population who primarily works from home, and the county-level fraction of the population that has lived in the same residence for more than one year. These thresholds range from 11% (county-level fraction of the population living in the same residence for more than one year) to 260% (county-level fraction of workers commuting by public transportation). These transformation thresholds have two interpretations based on the normativity of the risk factor and the sign of the transformation.

**Transformation vs. degradation**. We define transformation as a change in the risk factors of a county large enough that the county is now most similar to an alternate set of peers from its baseline. However, a substantial change in a risk factor could occur such that transformation has negative outcomes. Risks factors that are normatively good or neutral with a positive threshold represent a target for policymakers and decision-makers; we refer to these as 'positive transformation' or simply transformation throughout this paper. For example, in Bay County, a positive increase in the fraction of the population who commutes by public transportation—a normatively positive risk factor for improving the sustainability of a community[54] —will lead to positive transformation (Fig. 3a). This is in line with

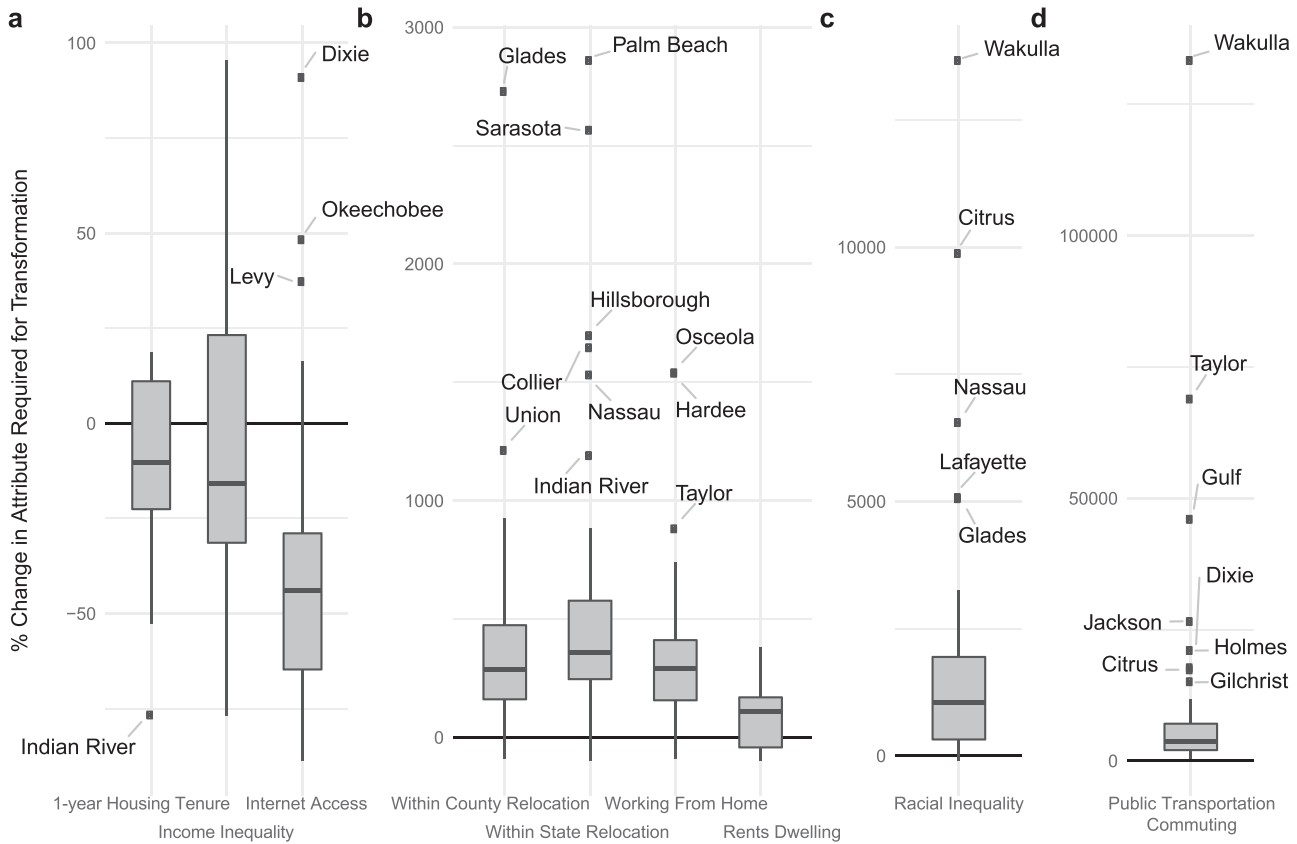

**Fig. 4 Risk factor change required to elicit transformation.** Each boxplot shows the percentage change in each risk factor required for a county to be mapped to an alternative node in the CCN, across all counties in the case study of Florida. Boxes represent 25th to 75th percentiles (the IQR), horizontal black lines are means, whiskers show 1.5 times the IQR, and points are outlying. (**a**) through (**d**) have increasing degrees of change required for transformation. Risk factors with no possibility of transformation are listed in Table 2. Note the shift in *y*-axis values moving from left to right.

previous work which has established the sustainability and resilience benefits of access to public transportation due to improving public health outcomes and providing equitable community connectivity[55,56]. Conversely, risk factors which are normatively negative, with negative transformation thresholds, serve as reduction targets.

Alternatively, risk factors which are normatively negative but with positive transformation thresholds indicate potential for negative transformation or degradation. Degradation indicators signal the risk of vulnerable communities' transition to worse outcomes outcomes. In Bay County, racial inequality and income inequality are both normatively negative risk factors which have positive transformation thresholds (Fig. 3a). A 29% increase in income inequality as measured by the Gini Index, for example, would lead to a negative transformation/degradation and would place Bay County among the highest levels of income inequality in Florida. As inequality in socioeconomic status is a key contributor to vulnerability[57], this threshold outlines the relative degradation risk faced by Bay County as a result of changes in income inequality.

**Triggering transformation.** Comparing the transformation thresholds for all risk factors and counties in Florida (Fig. 4) with the relative importance of risk factors identified as key contributors to recovery (Fig. 1), we find discrepancies in insights provided by the recovery model and CCN (Table 1). The risk factors deemed as transformation catalysts, i.e., those with the lowest transformation thresholds, are the county-level fraction of households who have lived in their current residence for over one

year (10.3%), county-level income inequality (15.8%), and county-level internet access (43.9%). However, the ranking of these three risk factors, in terms of their contribution to recovery, are 17th, 11th, and 18th, respectively (Fig. 1 a, Table 1).

Length of residence in a disaster-prone region is associated with decreased likelihoods of evacuation from major hurricanes and reduced perceptions of risk[58,59]. While the links between risk perception and community resilience are still being understood[60], we believe the importance of this risk factor in positively contributing to transformation comes from the place-based nature of community resilience and the social capital built with increased length of residence. Income inequality has also been tightly linked to disaster outcomes; having been identified as both a consequence of major disasters[61], and a driver of more severe disaster outcomes[62], and individual evacuation behavior[63]. Access to communication technology has also been linked to improved community resilience in previous work[64,65].

**Resilience traps.** The term trap is used in many instances to describe feedback loops in which governance and interventions designed to rectify a larger societal problem contribute or exacerbate the problem, such as poverty traps in which individuals are held in impoverished conditions by external forces[66], and rigidity traps when institutions and systems become self-reinforcing and inflexible[67]. Resilience traps can occur by an incomplete translation of resilience concepts to operational models[68]. Specifically, risk factors which are associated with recovery but do not allow for any possibility of positive transformation are defined as resilience traps in this paper.

**Table 1 CCN importance and engineering resilience importance.**

| Risk factor | CCN transformation threshold ranking (Fig. 3a) | Recovery importance ranking (Fig. 1a) |
|---|---|---|
| 1-year housing tenure | 1 | 17 |
| Income inequality | 2 | 11 |
| Internet access | 3 | 18 |
| Within county relocation | 4 | 16 |
| Within state relocation | 5 | 1 |
| Working from home | 6 | 4 |
| Rents dwelling | 7 | 12 |
| Racial inequality | 8 | 20 |
| Public transportation commuting | 9 | 8 |

Values in the left column represent the importance of the risk factor in contributing to transformation, with lower values having a smaller transformation threshold. Values in the right column are the importance of each risk factor as determined by their contribution to predicting community recovery.

**Table 2 Recovery importance for non-transformation risk factors.**

| Risk factor | Recovery importance ranking (Fig. 1a) |
|---|---|
| Cellphone only | 4 |
| Income deficit | 5 |
| GED | 7 |
| Abroad relocation | 9 |
| Speaking API | 10 |
| Speaking Spanish | 12 |
| Bachelor's degree | 13 |
| Commuting alone | 16 |
| Walking commute | 17 |
| Income through interest | 20 |

The risk factors listed have no contribution to transformation and the importance values are the relative contribution of each risk factor to predicting community recovery.

We find that of the risk factors included in the CCN, 11 of the 20 (55%) have no potential for transformation in any county evaluated in Florida while 9 allow for transformation in at least one (Fig. 4a–d). Of those which allow for transformation, county-level income inequality has the smallest mean transformation threshold (6.25% across all counties), while the county-level fraction of the population commuting by public transportation is the largest, with a mean transformation threshold of 12,042%. For the risk factors which do not allow for transformation, their contribution to recovery is listed in Table 2. The risk factors which do not allow for transformation range from the 4th to 20th most important variables as determined by the recovery model (Table 2).

The discrepancy in the importance between factors contributing to rapid recovery and triggering transformation highlight the possibility of resilience traps when aiming to operationalize the resilience of communities; and the potential barriers imposed by current resilience paradigms. Short-sighted policies, interventions, and investments motivated by solely prioritizing the factors which are associated with rapid recovery can entrench untenable and non-sustainable aspects of the status quo[5] and inhibit transformation needed to promote a sustainable and resilient society.

More specific to the case study presented here, the State of Florida leverages many federal disaster relief and resilience programs. Federal disaster programs such as FEMA Public Assistance grants pay part of the cost of rebuilding a community's damaged infrastructure[64], with the state providing matching funds to local and governments for the remainder. These funds are distributed based on criteria determined by the state including the requesting community's demographics (population size, poverty rate, unemployment rate), storm impacts (number of storms, severity of impact, non-FEMA reimbursable expenditures, additional hurricane recovery/mitigation funding granted, frequency of prior disasters, other assistance available), and the current revenue capacity of the requesting local government and the state. Updating these criteria to include risk factors which contribute to recovery and transformation—for example 1-year housing tenure and income inequality—could provide better long-term resilience outcomes by promoting investment in communities with the potential to both recover from disasters and transform.

## Discussion

Here, we highlight the importance of accounting for both recovery and transformation aspects of resilience, by integrating the results of the CCN and 'engineering resilience' (aka recovery modeling) approaches, and find that 55% of key predictors of recovery are potential resilience traps. For example, Income Deficit—a measure of poverty—is one risk factor which does not contribute to transformation but is positively associated with recovery. Income deficit quantifies the cumulative amount below the poverty line for all impoverished households the county[46]. This shows not just the number of households below the poverty line, but the degree of poverty experienced. We note that deficit is a negative value so higher values of it correspond with less cumulative poverty. Poverty contributes to increased disaster vulnerability as well as reduced capacity to cope with and recover from disasters[69–71]. In other words, poverty is a key driver of vulnerability to disasters, promoting dwelling in at-risk areas; disasters, in turn significantly increase poverty to the point of significantly shrinking or eliminating coping capacity[72]. Here, identifying poverty as a resilience trap likely points to the historical status of state and federal disaster aid in the US which addresses the disparate impacts of disasters across income classes through post-disaster relief as opposed to through poverty mitigation[33,34].

Income Through Interest—measured as the fraction of households receiving income through interest, dividends, or net rental income[46]—is another risk factor which does not contribute to transformation. Identifying this risk factor as a potential resilience trap should not come as a surprise, given the fact that disasters create permanent increases in rent in affected areas, while wealthy households expand their post-disaster real estate holdings[73]. Moreover, federal and state disaster relief policies have historically been based on measures of wealth and assets[64]. Many low-income households do not qualify for FEMA's disaster loans, and funding from HUD could take months or even years to reach impacted families[74].

Additionally, the Commuting Alone risk factor—the fraction of a county who primarily commutes alone by car—is negatively associated with engineering resilience, while the fraction of a county who commutes by walking (Walking Commute) is positively associated with engineering resilience. This confirms previous results which show the prioritization of post-disaster recovery favors more densely-populated downtown areas as opposed to suburban or rural areas[43]. Finally, we note that two education-based risk factors—the fraction of a county with bachelor's degrees and GEDs as their highest degrees earned (Bachelor's Degree, and GED)—are negatively associated with engineering resilience. This is attributable to the proximity of the

landfall location of Hurricane Michael to Leon County, FL. Leon County is home to Tallahassee and Florida State University both contributing to the county having the second highest proportion of Bachelor's earners in the state. This highlights the placed-based nature of resilience, where analyzing local data are necessary for gaining nuanced understanding of community's resilience.

Minority groups are frequently identified as vulnerable to disaster impacts[75,76], however in our case study the fraction of communities speaking primarily Spanish (Speaking Spanish) or Asian and Pacific Island languages (Speaking API) are both positively associated with community recovery, once socio-eonomic factors are accounted for. This indicates the presence of community organization and cohesion which are strongly associated with positive recovery outcomes in groups with strong bodies of shared cultural experiences[77,78]. Florida—particularly counties in South Florida—have the highest Hispanic and Latino population of any state in the southeastern US[46], highlighting the importance of conducting place-based analyses when developing operational models of resilience to inform decisions and policies.

As is the case with all of the aforementioned risk factors, policy decisions and interventions based recovery-oriented risk factors can neglect the transformative aspects of resilience which are required for long-term sustainability. These echos qualitative analyses of resilience-oriented policies which have found that the unilateral emphasis on restoring the status quo in engineering resilience models engenders norms and policies which inhibit the ability of communities to transform[6,79–81]. Transformation as a process within communities is critical when a current system is untenable[82], and thus will be vital for improving the sustainability of future communities[8,83]. We highlight the difference between the risk factors identified as important contributors to recovery versus transformation to emphasize the potential shortcomings of having a solely recovery-focused conceptualization of resilience and neglecting the importance of transformation in addition to post-disaster recovery.

By developing quantitative methods to assess the ability of communities to transform, we aim to shrink the gap between conceptual and operational models of resilience. We find that shifts in only a subset of risk factors allow for transformation within a community and within those, certain risk factors will lead to positive transformation if they are improved while others will lead to negative transformation if they deteriorate. Furthermore, we identify resilience traps in which existing, recovery-focused models place importance on risk factors which our model does not identify as important for transformation. We note that the choice of a county-level analysis was driven by power-outage data availability, and that there may be a significant distribution of individual risk factors within each county. This likely underestimates the impact on lower attainment households within each county who may be more vulnerable to disasters. The CCN methodology is scale agnostic and adaptable to higher-resolution demographic data such as census tracts or even blocks when appropriately scaled power-outage data becomes available. Similarly, this allows for the systems analyzed and scale of analysis to be tailored to the scale and importance needed for effective decision-making. In this way, decision and policymakers can evaluate the level of transformation achievable through implementation policies and interventions to promote sustainable and resilient lifestyles, economies, and societies.

## Methods

**Community risk factors**. We select 96 county-level variables to describe communities in the case study of Florida. The initial pool of variables are drawn from the American Community Survey[46,84] and Yale Program on Climate Change Communication[47]. The variables describe the sociodemographic, economic, housing, mobility, and environmental opinions for every county for the period of

time surrounding Hurricane Michael. A full list of included variable names and sources are listed in Supplementary Table 1.

**Storm exposure**. Storm exposure data is taken from the US National Centers for Environmental Information's Storm Events Database[85]. County-level exposure is included as a binary variable, labeled as true if the county is included in the Storm Events Database for Hurricane Michael, false otherwise. We also include a measure of distance to the storm center as a continuous variable. Distance is measured as the minimum distance between Hurricane Michael's center and the mean population center of each county, calculated with the R package STORMWINDMODEL[86], with a maximum distance of 1000 miles. We control for hazard exposure by including these two variables as covariates in the recovery model for the purpose of identifying the contribution of each risk factor to the recovery of the grid.

**System resilience: restored access to electricity**. To understand how risk factors contribute to 'engineering resilience', we measure the performance of the Florida electric power grid as impacted by 2018 Hurricane Michael. County-level power outages are taken from outage reports for the Florida Division of Emergency Management for October 10th through November 9th, 2018?. For each of the 67 counties in Florida, the Division of Emergency Management publishes the number of customers without power approximately every 3 h. At a time $t$, $Q(t)$ is the fraction of the county with access to power and represents the service level of the power system. We leverage a formal quantification of engineering resilience for a given county[23,87]. Resilience for a county, $R_{county}$ is the area under the service level curve, $Q(t)$ from the time of first disruption $t_0$ to the time when all outages are restored $t_f$ scaled by the difference between $t_f$ and $t_0$. $R_{county}$ is defined as

$$R_{county} = \frac{\int_{t_0}^{t_f} Q(t)}{|t_f - t_0|} \quad (1)$$

In this way, a county which lost all power immediately and remained so until it was recovery would have a resilience value of 0 and one with no disruption would have a resilience value of 1. Examples of the calculated resilience along with visual descriptions of $R$, $t_0$, $t_f$, and $Q(t)$ are shown in Supplementary Fig. 1a, b. Calculated resilience values for Hurricane Michael are given in Supplementary Table 2 and the repository linked in the Data Availability Statement.

**Engineering resilience model**. To identify the risk factors which contribute to 'engineering resilience', we utilize a predictive modeling paradigm. Predictive modeling aims to find a function, $y = \hat{F}(X)$ which maps inputs ($X$) to outputs ($y$) so as to minimize a measure of the distance between the predicted values and true values. Here, $y$ is the county-level resilience of the power grid and $X$ are the community risk factors, and $R^2$ and RMSE (Root Mean Square Error, Eq. (2)) are used as measures of distance. RMSE is defined as ins

$$RMSE = \sqrt{\frac{\sum_{i=1}^{n}(y_i - \hat{y}_i)^2}{n}} \quad (2)$$

Here, $n$ is the total number of observations in the test dataset, $y_i$ is the $i$th actual value of the response variable, and $\hat{y}_i$ is the response estimated by the model trained on the test data and evaluated on the test data. We train five model classes: linear models[88], generalized linear models[89], Random-Forest models[90], and Bayesian Additive Regression Trees[91] —all implemented in R[88]. Selecting model classes based on minimizing prediction error, however, can lead to overfit models in which the prediction error is reduced at the expense of generalization to non-training observations. To counteract this, we perform a 5-fold cross-validation procedure in which data is partitioned into 5 roughly equivalently sized folds[92]. Each fold—corresponding to approximately 20% of the data—is removed from the dataset, while remaining 4 fold are utilized to train the statistical models. The withheld fold (the test data) is then utilized to evaluate the out-of-sample predictive quality of the model. Out of sample RMSE and $R^2$ are shown in Supplementary Fig. 2.

Based on out of sample performance measures, we select a random-forest model to relate community risk factors to system resilience. Random forest is a tree-based, non-parametric statistical model[93]. To predict response values, the random-forest algorithm builds $B$ decision trees[94] on random subsets of the data. The data used in the tree creation are called the in the bag data, and the data not used is the out of the bag or OOB data. The random-forest algorithm averages the output over $B$ trees to create a final estimate of the predicted variable, $\hat{f}(x)$ such that

$$\hat{f}(x) = \frac{1}{B}\sum_{b=1}^{B} T_b(x) \quad (3)$$

Our final model is trained with 500 trees, an `mtry` value of 13 based on hyperparameter recommendations from[90].

**Variable selection algorithm**. To extract the importance using this trained model, we investigate the relative importance of community risk factors using the random-forest-based, three-step variable selection process VSURF[95,96] to determine which community risk factors most greatly contribute to single-equilibrium system resilience. VSURF, or Variable Selection Using Random Forest is an algorithmic

**Box 1 | Contrastive Community Network**

1: $r_i$ is a risk factor, where $r_i \in [R_i^{min}, R_i^{max}]$ and $X = \{r_i\} \, \forall \, i$
2: $c$ is a county where $c = \{1, \ldots, C\}$
3: $N(S, c)$ is the node of the self-organized map, $S$ which county $c$ maps to
4: $n_i$ is an arbitrary node $i$ in the SOM
5: $t_{r,c}$ is the temporal trajectory length for factor $r$ and county $c$, and $T = [t_{r,c}] \, \forall \, r, c$
6: $|n_i, n_j|_S$ as the euclidean distance in the SOM $S$ between nodes $i$ and $j$
7: Select predictive model $F$ using cross-validation
8: Select important features, $x$ using variable selection on $F$
9: Train SOM, $S = \text{SOM}(X')$
10: Tune SOM hyperparameters to minimize mean distance from county to node
11: **for all** $r \in X$ **do**
12: **for all** $c \in C$ **do**
13: $n_0 = N(S, c)$
14: $x_{r,c} = R^{min}$
15: **while** $x_{r,c} \in [R_i^{min}, R_i^{max}]$ **do**
16: $x'_{r,c} = x_{r,c} + \delta$
17: $S' = \text{SOM}(X')$
18: $n_1 = N(S', c)$
19: **If** $n_0! = n_1$ **then**
20: $t_{r,c} = |n_0, n_1|_{S'}$
21: **end if**
22: **end while**
23: **end for**
24: **end for**

CCN algorithm to return the set of temporal trajectories for each county and risk factor. In summary, the algorithm trains a self-organized map based on pre-selected community features, then systematically perturbs the values of the risk factors and remaps the counties to the SOM using the perturbed risk factors to determine if reconfiguration occurs. The final SOM topology used in the CCN in this case study is a 5 by 8 node hexagonal, toroidal grid.

process for selecting the importance of variables from random-forest models which aims to simultaneously find variables most related to the response for the purposes of interpretation, and to do this with the smallest set of variables possible[96]. The importance of a variable, $j$, in a random-forest model—denoted $VI(X^j)$—is computed by permuting variables to determine their sensitivity to the calculated error. Formally, errorOOB is the RMSE of a single tree on the data which was not used to construct it. For the variable $j$, $X^j$ is perturbed and the error calculated on the perturbed dataset, called $\widetilde{\text{error}}\text{OOB}$. The importance of the variable, then, is denoted as

$$VI(X^j) = \frac{1}{B} \sum_{b=1}^{B} (\widetilde{\text{error}}\text{OOB}_t^j - \text{errorOOB}_t) \qquad (4)$$

VI for each variable is shown in Fig. 1. The VSURF procedure begins by calculating VI for every variable included in the model, and sorting them in decreasing order of importance. Those below a threshold, chosen to be 2.95e−5 in our procedure, are removed. A series of random-forest models are then created with the step-wise addition of variables in descending order of importance until the mean errorOOB decreases by less than a pre-defined threshold. We defined these thresholds apriori and select the risk variables which meet the importance criteria, resulting in 20 risk factors.

**Contrastive community networks**. To develop contrastive community networks, we utilize Self-Organized Maps (SOM)[48,97,98]. SOMs are an unsupervised learning algorithm, based on artificial neural networks, for producing a low-dimensional, nonlinear representations of complex high-dimensional data[48]. SOM models are a graph of adjacent vertices in which each element in high dimensions is mapped to a node in the network. The process of assigning input data to nodes is done iteratively through a competitive learning process. The result is a graph (Fig. 2 a) which preserves the vectorial topology of the input data where closer nodes (called map units) within the map have higher similarity in the original input variables.

SOM models have been previously utilized for understanding the similarity between items in high-dimensional space without imposing assumptions on the structure of the data[99], and when looking for trends in spatiotemporal data relating to community and urban change[51,52,100].

What follows is a description of the SOM training process developed by Kohonen[48], and implemented in R[88,97]. For a fixed number of nodes (or map dimension), the training process assigns weights to each risk factor of the input data at each node in the map. In our experiments, 40 nodes were selected with 6 connections between neighbors based on SOM size heuristics[101], and confirmed by empirically observing the distance between nodes (Supplementary Fig. S4).

This creates the initial mapping between input space (original data) and output space (the SOM). The weights between nodes are initially assigned at random, then

a random input data point is selected. The winning map node—defined as the node with mean input data which is closest to the selected point—is selected. The weights between winning node and all others are updated by a value $\Delta w_{j,i}$ based on the number of iterations and the mean risk factor values of nodes within the selected node's topological neighborhood $T$. Eq. (5) shows the updating procedure of $\Delta w_{j,i}$

$$\Delta w_{j,i} = \eta(t) * T_{j,I(x)}(t) * (x_i - w_{j,i}) \qquad \text{for all} \qquad i, j \qquad (5)$$

where $i$ and $j$ refers to different neurons, $x_i$ is the value of the input data for node $i$, $t$ refers to iteration number, $I(x)$ refers to the winning neuron, and $w_{i,j}$ is the weight between node $i$ and $j$. The learning rate as a function of iteration is $\eta(t)$, where

$$\eta(t) = \eta_0 \exp(-t/\tau_n) \qquad (6)$$

and $\eta$ decreases with $t$ and based on a pre-assigned hyperparameter $\tau_n$, chosen in our experiments to be 0.05 based on previous empirical studies[97]. The topological neighborhood, $T$, defines how many neighboring nodes contribute to updating the learning rate of the selected node and is defined where

$$T_{j,I(x)}(t) = \exp(-S_{j,I(x)}^2 / 2\sigma(t)^2) \qquad (7)$$

and $S_{j,i}$ is the distance between weights such that $S_{j,i} = ||w_j - w_i||$ and $\sigma(t) = \sigma_0 \exp(-t/\tau_0)$, which shrinks the neighborhood size over successive iterations as well. This process of updating node weights is repeated for every input data point over a fixed number of iterations, chosen to be 10000 in our experiments based on empirically observing convergence of the distances between nodes.

We utilize the SOM algorithm as the basis for developing a Contrastive Community Network (CCN). The details of the CCN procedure are shown in Box 1, and described in summary here. Input variables for the CCN are the community risk factors, $r$, selected as important in the VSURF procedure for each county in Florida with storm exposure variables removed so as to compare communities on the basis of their structure rather than their hazard exposure. Risk factors are scaled to a standard deviation of 1 with mean 0 to facilitate the integration of input data of different magnitudes into the training of the SOM in line with previous empirical studies[97].

For each county, $c$, and each risk factor $r$, the initial node the county is mapped to in the SOM, $n_0$ is recorded. The value of the risk factor for the given county, $x_{r,c}$ is perturbed in increments of 0.01 ($\delta$ in Box 1) which is in units of standard deviation of each risk factor. Each risk factor is perturbed starting from its lower limit, $R_i^{min}$ to its upper limit, $R_i^{max}$.

For risk factors which implicitly have lower and/or upper limits based on the way they are calculated—like county-level fractions of the population or income inequality which are defined on the range [0, 1]—we scale the limits in the same way as the input data and utilize the scaled values as limits to the perturbation of each risk factor. For risk

factors without explicit limits—such as income deficit—we perturb values within a range of 1.5 times the minimum and maximum risk factor observed across the counties.

At each perturbation iteration, the perturbed risk factor $x'_{r,c}$ is included in the set of all risk factors across all counties, and an updated SOM, $S'$ is calculated. The node the county is mapped to with the updated risk factor values, $n_1$, is then compared against $n_0$. If the new node, $n_1$ is different than the original node $n_0$, the euclidean distance between them is denoted $t_{r,c}$ which represents the length of the transformation trajectory, and $x'_{r,c}$ at the value of the change is the transformation threshold.

Computing multiple SOMs with alternative input data and has previously been utilized to understand how high-dimensional data about the makeup of communities transform over time[51,52,100]. As neighboring nodes in the CCN are of higher similarity than those farther apart, a county being re-mapped to a node farther away indicates a greater degree of transformation; thus the length of the transformation trajectory represents the magnitude of reorganization as a result of the change in the community risk factor. This process is outlined in detail in the algorithm in Box 1.

**Reporting summary**. Further information on research design is available in the Nature Research Reporting Summary linked to this article.

## Data availability
Data not provided in supplementary materials or repositories are available upon reasonable request to B.R. Any data not provided in the repositories or supplementary materials are available through public repositories through the cited sources.

## Code availability
The transformation and resilience threshold data generated in this study have been deposited in the repository linked here https://doi.org/10.5281/zenodo.5591110. Census data and data from the Yale Program on Climate Change Communication are publicly available but not shared here due to their size. Processed data used for our models are shared in the linked repository.

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

## Acknowledgements

This work was supported by the National Science Foundation Grants CMMI-2000140 (R.N.) and CMMI-1826161 (R.N.)

## Author contributions

B.R. and R.N. came up with the idea and designed the study; B.R. conducted the analysis and made the figures. B.R. and R.N. drafted, reviewed, and edited the manuscript.

## Competing interests

The authors declare no competing interests.
