## [Peer Review File · Nature Communications]

Reviewers' Comments:

Reviewer #1:

Remarks to the Author:

The paper raises an important overall argument that too much emphasis on resilience is not necessarily a correct strategy by default and can prove counter-productive. Having said that, the paper does not follow this line of thought and a large part of the paper stays within the model and its results without fitting the earlier argument into the context of Hurricane Michael specifically, as well as its applicability to other similar challenges. Specific comments are provided below:

1. From the abstract, last line: maladaptation and disaster policies which emphasize resilience traps are not discussed in the paper
2. There is no discussion of the nature of the disaster itself. Hurricane Michael- why is this a good case? How common are hurricanes in Florida, was a substantial power outage seen for the first time? How long was the outage, how bad was it? Why is restored access to electricity the only chosen indicator of recovery?
3. Placement of sections need to be re-worked and show flow more logically. Methods section comes after discussion and conclusion and section 1 is already talking about the analysis. Fig 1 and 2 must come later in the paper.
4. Shortlisting of the 20 indicators from the longer list in the appendix is not clear. What is the relation of these to recovery from power outage, how and why were these selected? Similarly, Table 1 and 2, how were these risk factors identified- basis of transformation thresholds? Perhaps it can be made clearer.
5. A major issue with the paper is that the argument for transformation itself and what it looks like in the case chosen is not clear at all. Why is recovery itself not good enough? Why is transformation needed in this case? Other terms such as positive transformation, community transformation and maladaptation, maladaptive states, high sustainability future- what does these mean in the context of the case chosen?
6. The section on resilience traps is very generic. Especially statements such as short-sighted policies, non-sustainable aspects of status quo, what do these mean in the case context? What kind of disaster management policies does Florida already have in place? Do these fall short of what is needed? If yes, how? There is no background discussion of any of these.
7. Last 2 paras on Page 6- this is somewhat of a good summary or recommendation being derived from all the analysis. Authors should do away with any generic statements that are not backed by facts relevant to the case or emerging from the model itself.

Reviewer #2:

Remarks to the Author:

This topic presented in the manuscript is especially salient and timely: the driving factors for engineering resilience may inhibit transformation. The reviewer greatly appreciates the authors' work in exploring the driving factors of transformation and resilience. The reviewer also appreciates the authors' comments about the implications of this on mitigation and policy. However, the reviewer has some concerns regarding the statistical analysis done in this work (see specific comments below). The reviewer recommends a major review to address the comments and move toward publication.

Specific comments:

- 1) The reviewer's main concern is a general question/concern regarding transformation. The authors seem to be making the argument that we should be putting increased efforts into transformation instead of overemphasizing engineering resilience. However, the authors have defined transformation using a CCN with five different clusters that is hard to explain. Engineering resilience, although limiting, is presented in a very explainable way (electricity restoration). The reviewer is concerned that it will be hard to promote policy and mitigation on a metric that is not defined in an explainable way. In other words, it is difficult to tell what "transforming" from cluster B to cluster C is so why would I be motivated to use mitigation efforts to do that?
- 2) How did the authors choose to pick 20 of the most important variables for engineering resilience? Why not 10 or 5?
- 3) The authors note that they have controlled for exposure in the case study, but it is unclear to

the reviewer how this was done.

4) The authors seem to include every county in Florida for the case study. Did every county experience electricity loss due to Hurricane Michael? If not, it is unclear to the reviewer why these counties were left in the case study.

5) A county seems very large and the distribution of individual indicators within the county will be large. The reviewer is worried the averages presented in the study will not be representative of the lower attainment households in the county. This is especially salient because these may be the households being represented in the engineering resilience factor (i.e., those households may be the ones without electricity after the defined recovery period). It seems that the Kolm's inequality tries to measure this on a census tract level but other variables like this do not seem to be in the data set.

6) The reviewer does not understand Figure 2a and 2b. For Figure 2a, the distributions of the four variables in each hexagon are highly variable within the different clusters (colors). The reviewer understands that this is expected since the CCN is based on 20 different variables but the reviewer does not understand the message the authors are trying to convey with Figure 2a. With Figure 2b, the reviewer does not understand what the black dots in each of the hexagons represent. The reviewer understands that the colors represent different clusters identified by the CCN (Figure 2c).

7) Is there any scale to the clusters? I.e., is it better to be in cluster B than to be in cluster C? If so, how is that ordering conducted?

8) Since transformation is defined by moving from cluster to another using the CCN, it is important that this transformation means something. However, this explain-ability is going to be difficult given that the CCN is a deep learning technique. These generally have low explain-ability. Can the authors provide a bit more insight as to why they believe this a good measure for transformation? Moreover, based on the reviewer's understanding of the algorithm, the authors must first define the number of classes (or clusters) the CCN should have. How do the author's findings change with this parameter?

9) The authors discuss different ranges of perturbation to relate to mitigation. Are these perturbations realistic enough to provide some information about mitigation actions?

10) The scaling used works well for variables which are normally distributed. Some of the variables used are known to be skewed high. Does this cause any issues?

11) The final hyperparameters used for the statistical models does not seem to be in the manuscript.

Response to Reviewer & Editor Comments for the paper entitled *Overemphasis on recovery inhibits community transformation and creates resilience traps*

We thank the reviewers and editors for taking the time to read and review this manuscript as well as for their constructive comments and feedback. We have carefully revised the manuscript in line with the feedback from the reviewers, and have added text (highlighted in blue) and figures to the revised version of the manuscript. In summary, we have made substantial revisions to the manuscript by providing a more rigorous definition of transformation as well as including a strong rationale to justify our choice of case study. Moreover, we have included additional discussion to better ground our analytical findings related to the case study presented and added a substantial discussion of how risk factors identified by our CCN methodology can be directly implemented in Florida's disaster relief policies. The revisions are outlined below with a point-by-point response to each reviewer's comments. Please note that reviewers' comments are in italics while our answers are not. All text from the manuscript is colored blue, and citation numbers correspond to the reference section of this document.

Answers to Reviewer 1

Comment R1.1 *The paper raises an important overall argument that too much emphasis on resilience is not necessarily a correct strategy by default and can prove counter-productive. Having said that, the paper does not follow this line of thought and a large part of the paper stays within the model and its results without fitting the earlier argument into the context of Hurricane Michael specifically, as well as its applicability to other similar challenges.*

Answer to R1.1 We appreciate this comment and the reviewer’s careful analysis of the manuscript. Below, we have addressed all of the comments and included the changes made to the manuscript.

Comment R1.2 *From the abstract, last line: maladaptation and disaster policies which emphasize resilience traps are not discussed in the paper*

Answer to R1.2 We appreciate the reviewer’s careful review of the manuscript. We have added a more rigorous definition of maladaptation and included an additional discussion of the impacts of resilience policy on maladaptation. Page 1, lines 32:

Moreover, these frameworks may be promoting *maladaptation*—defined as actions that are beneficial in the short-term but ultimately increase vulnerability to future disruptions (2, 6). There is evidence that current disaster policies based on engineering resilience paradigms—such as insurance and disaster relief assistance programs—exacerbate wealth inequality (3) and broaden the racial wealth gap in areas impacted by disasters (7) by reinforcing the status quo which exacerbates persistent vulnerability rather than enabling adaptation (6). Operationalizing resilience paradigms that incentivize not only recovery but also transformation will enable designing disaster policies and interventions which do not exacerbate vulnerabilities and inequities.

Comment R1.3 *There is no discussion of the nature of the disaster itself. Hurricane Michael—why is this a good case? How common are hurricanes in Florida, was a substantial power outage seen for the first time? How long was the outage, how bad was it? Why is restored access to electricity the only chosen indicator of recovery?*

Answer to R1.3 This is an excellent point which will help strengthen the applicability of the manuscript. We have added the following text to describe why hurricanes are a suitable case study for studying community resilience as well as why we specifically studied Hurricane Michael in the state of Florida. Page 2, line 45:

We quantify threshold effects and conduct tipping point analyses by estimating the degree of change needed in risk factors to cause transformation, using the 2018 Hurricane Michael in Florida as a case study. The focus on hurricanes was due to the severity of their impact on communities and their wide-reaching devastation, positioning them as significant stress tests of community resilience. According to the National Oceanic and Atmospheric Administration, hurricanes have caused more deaths and destruction than any other recorded climate disaster in U.S. history. Hurricane Michael was specifically chosen as a case study as it is the most powerful storm to make landfall in Florida since the state began publicly reporting county-level disaster impacts (1). As of 2018, 41% of hurricanes that have hit the US have made landfall in Florida (8), and accordingly, the state has a significant number of programs designed to foster resilience

and aid in the immediate recovery from hurricanes (1). We use power outages —available at a county level throughout the storm— as a proxy for community recovery, as power outages were widespread throughout the state, affecting communities’ access to all other critical resources such as food, water, transport, and hygiene. Our findings demonstrate that an overemphasis on recovery and not accounting for transformation can entrench *resilience traps*, where risk factors that are predictive of recovery inhibit positive transformation and perpetuate maladaptive states.

Comment R1.4 *Placement of sections need to be re-worked and show flow more logically. Methods section comes after discussion and conclusion and section 1 is already talking about the analysis. Fig 1 and 2 must come later in the paper.*

Answer to R1.4 We appreciate the feedback on the style and flow of the paper. We have carefully re-read the paper to ensure its logical flow and clarity of presentation. However, the order of the other sections –e.g. methods coming after the discussion and a discussion of the findings in the introduction– are based on the the author’s instruction guide by *Nature* which aims to make the study accessible to a wide range of audience/scholars.

Comment R1.5 *Shortlisting of the 20 indicators from the longer list in the appendix is not clear. What is the relation of these to recovery from power outage, how and why were these selected? Similarly, Table 1 and 2, how were these risk factors identified- basis of transformation thresholds? Perhaps it can be made clearer.*

Answer to R1.5 Thank you for the comment, we have added the text in blue below to the *Quantifying Transformation* section of the text describing how the final 20 indicators were selected, and have updated the captions in Tables 1 and 2 to better reflect the chosen variables. Page 3, line 68:

We select a subset of 20 risk factors which are most predictive of restored access to electricity (Fig. 1), using an ensemble-of-trees predictive model. Specifically, using an exhaustive search approach, our three-stage variable selection algorithm (see Methods) searches through all possible combination of input variables and identifies the *smallest subset* of risk factors that are most predictive of community recovery (see Methods). These risk factors represent what a data-driven approach to engineering resilience would identify as as most important for restoring access to critical services.

Moreover, the details of the variable selection algorithm is outlined in the Methods Section, under the ‘Variable Selection Algorithm’ subsection on page 9 of the manuscript.

Comment R1.6 *A major issue with the paper is that the argument for transformation itself and what it looks like in the case chosen is not clear at all. Why is recovery itself not good enough? Why is transformation needed in this case? Other terms such as positive transformation, community transformation and maladaptation, maladaptive states, high sustainability future- what does these mean in the context of the case chosen?*

Answer to R1.6 We truly appreciate this constructive feedback. We have significantly expanded the results and discussion sections to provide more clarifications on the terminology used to describe the CCN methodology and what these terms mean in the case study chosen. First, we included

text in the main body of the manuscript to provide a more thorough explanation of our definition of transformation. Page 4, line 95:

We use a county being mapped to a non-baseline node as an indicator of transformation. In other words, in the CCNs architecture, transformation occurs when a change in the risk factors of a county is large enough, such that the county’s position in the relational network shifts and it becomes more similar to an alternate set of peers from its baseline. By defining transformation relative to the peers of a county, we avoid imposing judgment about the magnitude of change in a risk factor significant enough to be identified as transformation.

Answer to R1.6 Additionally, in the section of the manuscript discussing the results in Bay County, FL, we have provided a more detailed definition of how our definition of transformation using CCNs complements recovery to justify why transformation is needed. Page 4, line 107:

Stated alternatively, improving these risk factors will enhance both the immediate disaster recovery of the county –as determined by the recovery model– and will potentially change the county’s similarity to its peers. Changes in the remaining 60% of the risk factors, however, can only improve the recovery of the community, and would not alter the underlying similarity of the county to its peers (i.e., not conducive to transformation).

Answer to R1.6 Similarly, we have added thorough definitions and examples for transformation thresholds. Page 4, line 112:

We conceptualize a transformation threshold as the magnitude of change in a risk factor required to trigger transformation. The aim is to jointly identify the risk factors within a community which are conducive to triggering transformation, and determine the degree of change needed in these risk factors for transformation to occur.

Answer to R1.6 And for positive transformation. Page 5 line 129:

Risk factors that are normatively good or neutral with a positive threshold represent a target for policymakers and decision-makers; we refer to these as ‘positive transformation’ or simply transformation throughout this paper.

Answer to R1.6 We removed the terms ‘community transformation’, ‘maladaptive states’, and ‘high sustainability future’ in an effort to use more precise language, and included a definition of ‘maladaptation’ in the introduction. Again, we appreciate the comment, and hope that these revisions have helped improve the focus, precision, and clarity of the manuscript.

Comment R1.7 *The section on resilience traps is very generic. Especially statements such as short-sighted policies, non-sustainable aspects of status quo, what do these mean in the case context? What kind of disaster management policies does Florida already have in place? Do these fall short of what is needed? If yes, how? There is no background discussion of any of these.*

Answer to R1.7 We appreciate the comment and have added specific details related to our case study in the resilience traps section. The added text is below. Page 6, line 176:

More specific to the case study presented here, the State of Florida leverages many federal disaster relief and resilience programs (1). Federal disaster programs such as FEMA Public Assistance grants pay part of the cost of rebuilding a community’s damaged infrastructure (5), with the state providing matching funds to local and governments for the remainder. These funds are distributed based on criteria determined by the state including: the requesting community’s demographics (population size, poverty rate, unemployment rate), storm impacts (number of storms, severity of impact, non-FEMA reimbursable expenditures, additional hurricane recovery/mitigation funding granted, frequency of prior disasters, other assistance available), and the current revenue capacity of the requesting local government and the state (1). Updating these criteria to include risk factors which contribute to recovery and transformation –for example 1-year housing tenure and income inequality– could provide better long-term resilience outcomes by promoting investment in communities with the potential to *both* recover from disasters and transform.

Comment R1.8 *Last 2 paragraphs on Page 6- this is somewhat of a good summary or recommendation being derived from all the analysis. Authors should do away with any generic statements that are not backed by facts relevant to the case or emerging from the model itself.*

Answer to R1.8 We appreciate the comment and have pruned the discussion to only include conclusions backed specifically by our analysis and model results.

Answers to Reviewer 2

Comment R2.1 *This topic presented in the manuscript is especially salient and timely: the driving factors for engineering resilience may inhibit transformation. The reviewer greatly appreciates the authors' work in exploring the driving factors of transformation and resilience. The reviewer also appreciates the authors' comments about the implications of this on mitigation and policy. However, the reviewer has some concerns regarding the statistical analysis done in this work (see specific comments below). The reviewer recommends a major review to address the comments and move toward publication.*

Answer to R2.1 We appreciate the reviewer's positive feedback as well as comment related to the explainability of the work to improve the applicability and usefulness of this work.

Comment R2.2 *The reviewer's main concern is a general question/concern regarding transformation. The authors seem to be making the argument that we should be putting increased efforts into transformation instead of overemphasizing engineering resilience. However, the authors have defined transformation using a CCN with five different clusters that is hard to explain. Engineering resilience, although limiting, is presented in a very explainable way (electricity restoration). The reviewer is concerned that it will be hard to promote policy and mitigation on a metric that is not defined in an explainable way. In other words, it is difficult to tell what "transforming" from cluster B to cluster C is so why would I be motivated to use mitigation efforts to do that?*

Answer to R2.2 We have significantly revised the text to provide a better definition of transformation as quantified by CCNs, and to create a more approachable explanation of this definition using the chosen case study. We have added the following text to the *Transformation Trajectories and Thresholds* section to provide a more thorough definition of transformation. Page 4, line 95:

We use a county being mapped to a non-baseline node as an indicator of transformation. In other words, in the CCNs architecture, transformation occurs when a change in the risk factors of a county is large enough, such that the county's position in the relational network shifts and it becomes more similar to an alternate set of peers from its baseline. By defining transformation relative to the peers of a county, we avoid imposing judgment about the magnitude of change in a risk factor significant enough to be identified as transformation.

Answer to R2.2 Similarly, we have added the following text describing transformation thresholds. Page 4, line 112:

We conceptualize a transformation threshold as the magnitude of change in a risk factor required to trigger transformation. The aim is to jointly identify the risk factors within a community which are conducive to triggering transformation, and determine the degree of change needed in these risk factors for transformation to occur.

Answer to R2.2 And provided a more detailed definition of positive transformation. Page 4, line 129:

Risks factors that are normatively good or neutral with a positive threshold represent a target for policymakers and decision-makers; we refer to these as 'positive transformation' or simply transformation throughout this paper.

Answer to R2.2 Finally, we have added the following text to the discussion with the aim of emphasizing that transformation and recovery are complementary components of resilience:

We highlight the difference between the risk factors identified as important contributors to recovery versus transformation to emphasize the potential shortcomings of having a solely recovery-focused conceptualizations of resilience and neglecting the importance of transformation in addition to post-disaster recovery.

Comment R2.3 *How did the authors choose to pick 20 of the most important variables for engineering resilience? Why not 10 or 5?*

Answer to R2.3 We use an exhaustive search algorithm that permutes through all combinations of input variables to identify *the smallest subset* of variables needed for optimal prediction (such that the predictive performance is equivalent/better than a more saturated model (meaning a model with more predictor variables). Removing any of the 20 selected variables from the final model would lead to a statistically significant loss in accuracy. We have included more clarification on the variable selection algorithm as outlined below.

We select a subset of 20 risk factors which are most predictive of restored access to electricity (Fig. 1), using an ensemble-of-trees predictive model. Specifically, using an exhaustive search approach, our three-stage variable selection algorithm (see Methods) searches through all possible combination of input variables and identifies the *smallest subset* of risk factors that are most predictive of community recovery (see Methods). These risk factors represent what a data-driven approach to engineering resilience would identify as as most important for restoring access to critical services.

Moreover, the details of the variable selection algorithm is outlined in the Methods Section, under the ‘Variable Selection Algorithm’ subsection on page 9 of the manuscript.

Comment R2.4 *The authors note that they have controlled for exposure in the case study, but it is unclear to the reviewer how this was done.*

Answer to R2.4 We appreciate this comment. To clarify: we include two county-level variables representing exposure –a binary variable generated by the US Center for Environmental Information indicating whether a county was exposed or not, and a continuous variable calculating the minimum distance between the storm’s center and the county’s population center– as covariates in the initial recovery model which assess the contribution of the county-level risk factors toward engineering resilience. We have added the following text to the *Storm Exposure* part of the Methods Section. Page 8, line 252:

We control for hazard exposure by including these two variables as covariates in the recovery model for the purpose of identifying the contribution of each risk factor to the recovery of the grid.

Comment R2.5 *The authors seem to include every county in Florida for the case study. Did every county experience electricity loss due to Hurricane Michael? If not, it is unclear to the reviewer why these counties were left in the case study.*

Answer to R2.5 The reviewer is correct. We included every county in the case study, and just over 40% of the counties in the state experienced some form of power outage. We did this intentionally because we quantify a county’s transformation relative to other counties and thus needed to create a sizable and representative population to compare against. However, by statistically controlling for hazard exposure –rather than removing non-exposed counties– the model accurately accounts for the non-impacted counties.

Comment R2.6 *A county seems very large and the distribution of individual indicators within the county will be large. The reviewer is worried the averages presented in the study will not be representative of the lower attainment households in the county. This is especially salient because these may be the households being represented in the engineering resilience factor (i.e., those households may be the ones without electricity after the defined recovery period). It seems that the Kolm’s inequality tries to measure this on a census tract level but other variables like this do not seem to be in the data set.*

Answer to R2.6 This is an excellent point and we completely agree with the reviewer. However, unfortunately, we only have access to county-level power outage data. We agree that this is likely underestimating the impact of the disaster on lower attainment households in the county who are likely more vulnerable, and we have added the text below in the discussion to highlight this excellent point. Our use of the Kolm’s inequality measures was to create a county-level, aggregate indicator of the distribution of values within the county when appropriate; without sufficiently high-resolution power outage data we proceeded with our analysis at a county level.

We note that the choice of a county-level analysis was driven by power-outage data availability, and that there may be a significant distribution of individual risk factors within each county. This likely underestimates the impact on lower income households within each county who may be more vulnerable to disasters. The CCN methodology is scale agnostic and adaptable to higher-resolution demographic data such as census tracts or even blocks when appropriately scaled power outage data becomes available.

Comment R2.7 *The reviewer does not understand Figure 2a and 2b. For Figure 2a, the distributions of the four variables in each hexagon are highly variable within the different clusters (colors). The reviewer understands that this is expected since the CCN is based on 20 different variables but the reviewer does not understand the message the authors are trying to convey with Figure 2a. With Figure 2b, the reviewer does not understand what the black dots in each of the hexagons represent. The reviewer understands that the colors represent different clusters identified by the CCN (Figure 2c).*

Answer to R2.7 We appreciate the reviewer’s constructive feedback on the clarity of the figures. Figure 2a aims to highlight the variability in risk factors across the clusters, and to show that they vary in multiple dimensions simultaneously; we have updated the figure caption to reflect that. Similarly, as the reviewer has correctly interpreted, the colors in 2b correspond with 2c, and the dots show how the counties are mapped to each node. We have also updated the caption of the figure to improve clarity.

Comment R2.8 *Is there any scale to the clusters? I.e., is it better to be in cluster B than to be in cluster C? If so, how is that ordering conducted?*

Answer to R2.8 This is an excellent question. There is no ordinal scale associated with the clusters, as we quantify transformation in the relational network with respect to the degree of change needed in a risk factor such that a county is subsequently mapped to an alternative cluster. We do this to avoid making judgement calls about how much change in a risk factor is 'sufficient' for transformation. However, if an increase in a risk factor which is normatively positive (such as educational attainment) causes transformation, we label that *positive transformation*. We have clarified this by adding a significant amount of text to the sections describing transformation, as shown below. Page 4, line 112:

We conceptualize a transformation threshold as the magnitude of change in a risk factor required to trigger transformation. The aim is to jointly identify the risk factors within a community which are conducive to triggering transformation, and determine the degree of change needed in these risk factors for transformation to occur.

and on Page 5, line 129:

Risk factors that are normatively good or neutral with a positive threshold represent a target for policymakers and decision-makers; we refer to these as 'positive transformation' or simply transformation throughout this paper.

Comment R2.9 *Since transformation is defined by moving from cluster to another using the CCN, it is important that this transformation means something. However, this explain-ability is going to be difficult given that the CCN is a deep learning technique. These generally have low explain-ability. Can the authors provide a bit more insight as to why they believe this a good measure for transformation?*

Answer to R2.9 The reviewer raises a very important point regarding the trade-offs between model accuracy, complexity, and interpretability. We completely agree that these trade-offs have to be weighed carefully in light of the objective of the analysis. This is precisely the reason why we selected an ensemble-of-trees approach for the engineering resilience model. In other words, while the state-of-the-art deep learning approaches could potentially offer a slightly higher predictive accuracy, the black-box nature of the algorithms and the inability to do comprehensive and rigorous model inferencing rendered their use inappropriate for the objective of the analysis. On the other side, the tree-based approach, while less interpretable than fully parametric models, offered a significantly improved accuracy over (even the more flexible) parametric approaches.

Having said that, in the transformation modeling section of the analysis, the goal was to identify an efficient algorithm that could *objectively* characterize the trajectory of communities with respect to their peers without having to create arbitrarily (amorphous) complex indices (which is the prevalent approach today). In other words, we proposed the CCN as an alternative to the the more non-transparent ways of tracking community trajectory. In this methodology, the aim of the algorithm is to classify communities that are most similar to each other based on a wide suite of risk factors (without having to collapse the individual risk factors into a single measure/metric) such that the decision makers can simultaneously track communities' trajectory with respect to one another and at the same time be able to view the raw granular risk factors for all communities.

Comment R2.10 *Moreover, based on the reviewer's understanding of the algorithm, the authors must first define the number of classes (or clusters) the CCN should have. How do the author's*

findings change with this parameter?

Answer to R2.10 This is an excellent point, and the reviewer is correct in identifying that the number of classes is as an input. We utilize a sample-size heuristic developed by (9) to identify 40 nodes as a suitable number for the CCN. We subsequently validated that by looking at the mean distance between each county and its assigned node as a function of the number of nodes; we have now included this as Supplemental Figure S8.

Comment R2.11 *The authors discuss different ranges of perturbation to relate to mitigation. Are these perturbations realistic enough to provide some information about mitigation actions?*

Answer to R2.11 All perturbations are made within either (1) realistic upper and lower boundaries (e.g. counts do not drop below 0, fractions of a county do not exceed [0,1]), or (2) 1.5 times the observed minimum and maximum value for each risk factor. The focus of this work is identifying risk factors which contribute to transformation in any way, but not to recovery.

Comment R2.12 *The scaling used works well for variables which are normally distributed. Some of the variables used are known to be skewed high. Does this cause any issues?*

Answer to R2.12 This is a great observation. We have not attempted alternative variable scaling, but for the final set of 20 risk factors, only income deficit and population are not represented as fractions of a county and thus are the only risk factors not ranging between 0 and 1. Population and income deficit both have values in the thousands to millions. This discrepancy in magnitude is recommended to be addressed in Wehrens and Buydens (10) by scaling all variables to mean zero, standard deviation one.

Comment R2.13 *The final hyperparameters used for the statistical models does not seem to be in the manuscript.*

Answer to R2.13 We thank the reviewer for catching this. We have now included the parameters used in both the recovery model and the CCN in the Methods Section of the manuscript. Page 9 line 284:

Our final model is trained with 500 trees, an mtry value of 13 based on hyperparameter recommendations from (4).

and in the caption for Algorithm 1, as well as Supplemental Figure S8.

References

- [1] Hurricane michael - power outage data.
- [2] Jon Barnett and Saffron O’neill. Maladaptation, 2010.
- [3] Junia Howell and James R Elliott. Damages done: The longitudinal impacts of natural hazards on wealth inequality in the united states. *Social problems*, 66(3):448–467, 2019.
- [4] Andy Liaw and Matthew Weiner. Classification and regression by randomforest. *R News*, 2(3):18–22, 2002. URL <https://CRAN.R-project.org/doc/Rnews/>.
- [5] Mitigation Framework Leadership Group. *Draft Interagency Concept for Community Resilience Indicators and National-Level Measures*. Jun 2016. URL <https://www.fema.gov/media-library-data/1466085676217-a14e229a461adfa574a5d03041a6297c>.
- [6] Paul O’Hare, Iain White, and Angela Connelly. Insurance as maladaptation: Resilience and the ‘business as usual’paradox. *Environment and Planning C: Government and Policy*, 34(6): 1175–1193, 2016.
- [7] Kevin T Smiley, Junia Howell, and James R Elliott. Disasters, local organizations, and poverty in the usa, 1998 to 2015. *Population and Environment*, 40(2):115–135, 2018.
- [8] Adam B. Smith and Richard W. Katz. US billion-dollar weather and climate disasters: Data sources, trends, accuracy and biases. 67(2):387–410. ISSN 0921-030X, 1573-0840. doi: 10.1007/s11069-013-0566-5. URL <http://link.springer.com/10.1007/s11069-013-0566-5>.
- [9] Jing Tian, Michael H. Azarian, and Michael Pecht. *Anomaly Detection Using Self-Organizing Maps-Based K-Nearest Neighbor Algorithm*.
- [10] Ron Wehrens and Lutgarde M. C. Buydens. Self- and super-organizing maps in R: The kohonen package. *Journal of Statistical Software*, 21(5):1–19, 2007. doi: 10.18637/jss.v021.i05.

Reviewers' Comments:

Reviewer #1:

Remarks to the Author:

The authors have now responded to the review comments to satisfaction.

Reviewer #2:

Remarks to the Author:

The author's have adequately answered the reviewer's questions and the reviewer recommends publication with these changes.